# Physical Activity during COVID-19 Lockdown in Italy: A Systematic Review

**DOI:** 10.3390/ijerph18126416

**Published:** 2021-06-13

**Authors:** Luciana Zaccagni, Stefania Toselli, Davide Barbieri

**Affiliations:** 1Department of Neuroscience and Rehabilitation, University of Ferrara, 44121 Ferrara, Italy; luciana.zaccagni@unife.it (L.Z.); davide.barbieri@unife.it (D.B.); 2Center of Sport and Exercise Sciences, University of Ferrara, 44123 Ferrara, Italy; 3Department of Biomedical and Neuromotor Science, University of Bologna, 40126 Bologna, Italy

**Keywords:** pandemic, coronavirus, physical exercise, general health

## Abstract

The recent COVID-19 pandemic has imposed a general lockdown in Italy, one of the most affected countries at the beginning of the outbreak, between 9 March and 3 May 2020. As a consequence, Italian citizens were confined at home for almost two months, an unprecedented situation, which could have negative effects on both psychological and physical health. The aim of this study was to review the published papers concerning the effects of the lockdown on physical activity and the consequences on general health. As expected, most studies highlighted a significant reduction in the amount of performed physical activity compared to before lockdown, in both the general population and in individuals with chronic conditions. This fact had negative consequences on both general health, in terms of increased body mass, and on specific chronic conditions, especially obesity and neurological diseases.

## 1. Introduction

A new coronavirus (SARS-CoV-2), causing severe acute respiratory syndrome, was discovered at the end of 2019. The first cases were reported in the province of Wuhan, China. On the 30 January 2020, the Director-General of the World Health Organization (WHO) declared the SARS-CoV-2 outbreak a public health emergency of international concern, the WHO’s highest level of alarm. The associated syndrome was named coronavirus disease 2019, abbreviated as COVID-19, and it is currently a major global health issue. At the time of the submission there have been 147,539,302 confirmed cases of COVID-19, including 3,116,444 deaths.

Given the exponential growth in cases and deceased during February and early March 2020, and the risk of collapse of the national health system (in particular of intensive care units), the Italian government declared a general lockdown on 9 March 2020. This resolution remained in place until 3 May 2020, and implied that Italian citizens all over the country had to remain confined at home for almost two months, and could go out only for primary necessities, such as buying food or medicines, or seeking medical care. National policies imposed social distancing, the closure of schools and universities, and the suspension of any social event. All activities practiced in gyms, sports centers, and swimming pools were suspended. Additionally, jogging or walking in parks and cycling were prohibited. Essential activities (agriculture, biomedical manufactory, information and communication technology, medical care, energy production, and similar) were maintained, but when possible, work (for example administrative tasks and meetings) had to be performed from home, using internet connections and web conferencing tools. The imperative was “stay at home” to curb the spread of the virus.

As a consequence, a drastic reduction in the amount of performed individual physical activity (PA) was expected, which could have negative consequences on the population’s general health. It is well known that physical inactivity causes over 5 million deaths worldwide and represents damage to the economy of the public health systems. In particular, it could have a negative effect on glycemic control, which is especially dangerous for subjects with diabetes [1,2]. In obese patients, a further increase in body mass would worsen their condition. In cardiac patients, daily moderate PA is essential to reduce cardiovascular risk [3,4]. In patients with neurological disorders, exercising is of fundamental importance to control their conditions, in terms of both motor control [5,6] and cognitive impairment [7]. Furthermore, being locked at home implied a reduction in the exposure to open air and sunlight, which could undermine the immune system and its capability to react to the new viral infection. In particular, PA has immune benefits, especially in older adults [8] who are more at risk during the current pandemic. 

The aim of this paper was to perform a systematic review of the studies on PA carried out at home by the Italian population during the lockdown period, in order to evaluate the changes in lifestyle, whether sedentary or active, and the eventual consequences on general health. The underlying hypothesis is that home confinement has diminished the amount of PA, with negative consequences on individual health in general.

## 2. Materials and Methods

A systematic review of the studies investigating the PA practiced during the lockdown by Italian citizens was conducted in accordance with the *Preferred Reporting Items for Systematic Reviews and Meta-Analyses* (PRISMA) guidelines [9]. 

An electronic literature search was performed on the PubMed and Scopus online databases, using the following search string: (“physical activity” OR “physical exercise”) AND (COVID-19 OR lockdown) AND Italy. As inclusion criteria, we considered only articles in the English language and only studies performed on Italian citizens; concerning the publication type, we considered only research papers, excluding reviews, letters to editor, comments, editorials, and recommendations. The search was conducted till 31 December 2020. 

A total of 136 records were collected. Two reviewers (D.B. and L.Z.) selected the relevant studies separately on the basis of the titles and abstracts. Then, they independently reviewed the full text of the selected studies to decide on their final suitability according to the inclusion and exclusion criteria. In case of disagreement, the decision was made collegially with the contribution of a third investigator (S.T.).

After the exclusion of 92 articles, we included 23 full-text papers to critically evaluate the role of PA during the lockdown caused by COVID-19 in Italy. The following information was collected from each of the included articles: focus, study design, sample description (size, age, sex (% males)), method of PA-related data collection (type of PA assessment tool and type of survey), amount of PA (before and during lockdown, when reported), and main findings.

The assessment of the methodological quality of the selected studies was carried out using the Newcastle–Ottawa scale (NOS) [10], adapted for cross-sectional studies [11], independently by two reviewers. A third reviewer was available to resolve any disagreements. Each study was assessed with the following items: selection (representativeness of the sample, sample size, non-respondents, and ascertainment of the exposure), comparability, and outcome (assessment of the outcome and statistics). Scores range from 0 to 10, with higher scores indicating better quality research.

The whole process is described in Figure 1 by means of the standard PRISMA flowchart.

## 3. Results

Twenty-three published research papers that met inclusion criteria were selected and included in the review. The collected information of the included studies is summarized in Table 1.

All papers were observational studies. Specifically, the study by Predieri et al. [16] was longitudinal, while all others were cross-sectional.

Samples were different according to size (from a minimum of 24 participants in [20] to a maximum of 7847 in [27], age (from adolescents [16,17,34] to old adults [20], sex percentage (males vary from 16% in [24] to 62% in [13], and type (subjects with and without pathologies, students).

Ten studies (43.5%) were carried out on individuals with pathologies: six samples with Type 1 diabetes, one with neuromuscular disease, one with implantable cardioverter-defibrillator, one with Parkinson disease, and one with obesity. Studies with diabetes were concerned with glycemic control during lockdown. In one study [17], Type 1 diabetes patients continued to exercise regularly as before during the lockdown period without significant variations, even if they were confined at home, and were able to maintain good glycemic control. Notwithstanding the reduction of performed PA, three [14,15,16] of such studies found an improvement in glycemic control. Two studies [12,13] found a reduction in both glycemic control and PA.

Obese subjects worsened their condition [19]: body mass increased significantly and was associated with a reduction on PA. In addition, patients with a cardiovascular disease [20] reduced the amount of performed PA, which could increase their health-related risk.

Concerning neuromuscular diseases, one study [18] found a worsening of the patients’ condition, associated to a reduction in PA. One study [21] did not find significant changes before and during lockdown in the number of Parkinson’s disease patients who practiced PA. Nonetheless, those who declared a worsening of their condition reported a lower amount of daily PA compared to those who did not experience a worsening, highlighting the potential protective effect of PA. 

Four studies were carried out on students: one on high school [34] and three on university students [28,29,31]: all of them reported a reduction in PA and an increase in sedentary behavior. 

All studies but one used self-reported data collected during some form of interview (telemedicine visits or telephone contact) or by means of online questionnaires. Only one study [20] used smart devices to automatically and directly collect the data. Another study [23] used smart devices to measure the number of daily performed steps, but measurements were self-reported by study participants. Concerning the questionnaire used to assess the level of PA, only eight studies (34.8%) administered the validated IPAQ or IPAQ-SF; Assaloni et al. [12] administered the Godin Leisure Time Exercise Questionnaire and the remainders did not use validated questionnaires.

In Di Corrado et al. [25], the amount of individuals who began PA during lockdown was significantly greater than those who stopped. Di Renzo et al. [26] reported an increase in the percentage of individuals who trained five or more times a week. The studies which quantified the amount of performed PA found a relevant reduction during lockdown, compared to pre-lockdown.

The results of the assessment of the quality of studies are detailed in Table 2; the mean NOS score of the included studies was 5.04 (SD = 1.36; range 2–8).

## 4. Discussion

There is a lack of consistency in the findings of the selected studies, in terms of the amount of performed PA, even if a majority declared a significant reduction during lockdown compared to before, as expected. Overall, the two-month lockdown imposed as a consequence of the COVID-19 pandemic had a detrimental effect on general health in Italians, especially those with chronic conditions such as obesity and neurological diseases. Coping with such diseases has been a challenge, especially because of issues in supplies and lack of access to health facilities and healthcare providers. Overall, a reduction of PA was detected as a consequence of COVID-19 lockdown, with a worsening of health status.

An increase of PA was only reported by Di Corrado et al. [25] and Di Renzo [26]. The results of Di Renzo et al. [26] suggested that highly active people maintained or increased their PA level; Di Corrado et al. [25] reported an increase of PA beginners during lockdown. This may be that highly physically active people have reached an acknowledgement of healthy lifestyle which allows them to maintain it, including in critical conditions such as lockdown or confinement.

Studies on diabetes and glycemic control have shown contradictory results. Notwithstanding the diminished amount of performed PA reported by most patients, glycemic control improved, possibly because of a more regular lifestyle and daily timetable. It is also possible that the use of telemedicine (video and phone calls), which allows patients to conduct virtual visits, has been a useful support to manage this chronic disease [16]. Still, one study reported a reduction in glycemic control, and therefore the association between meal schedule and PA in the treatment of diabetes should be investigated further. 

The role of diet during lockdown was examined, especially in studies on obese and diabetic subjects; in particular, Pellegrini et al. [19], in their study on obese people, highlighted that even if all the patients received personalized nutritional advice, they reported many unhealthy dietary habits, such eating more, not paying attention to the healthiness of the consumed food, consuming more sweets, more snacks, more frozen/canned foods, and less fruit and vegetables than before. Moreover, Tornese et al. [17], in a study on diabetic adolescents, found a negative change in eating behavior.

Concerning neuromuscular diseases, the subjects who practiced PA have not shown signs of worsening. Schirinzi et al. [21] pointed out that higher educational level and a mild motor impairment are significant predictors of daily PA in Parkinson’s disease patients, since they seem to have a greater awareness of the importance of regular exercise and a more resilient response to COVID-19-related changes. Particular attention should be paid to involve more advanced, cognitively impaired, or uneducated patients who could be excluded from telecommunications.

The main limitation of most of the studies was that data were self-reported by the participants by means of a questionnaire, a web survey, or a phone interview. Only in one study [20] were all data collected automatically by means of electronic sensors. In another one [22], parts of the data were collected before lockdown by means of direct anthropometric measurements, while during lockdown, data were collected by means of a phone interview. The studies that reported the amount of PA are limited and heterogeneous, and this represents an obstacle in the interpretation of the results. 

Schirinzi et al. [21], in their study regarding the use of available technology-based tools to assist physical exercise and its implications in such a critical period, reported that levels of MET did not differ between users and the nonusers, probably due to the old age of the sample. However, in our opinion, an increased adoption of remote monitoring systems should be envisaged to keep the amount of PA performed by both patients and healthy subjects under direct control, avoiding the collection of self-reported—i.e., non-observed—data. Data on PA can also be provided by popular fitness monitoring cell phone applications. There are at least two main advantages in their adoption: (i) a more objective evaluation of both PA and biomedical parameters, and (ii) the possibility for physicians to avoid close contact with suspected or infected patients, thus diminishing their exposure to the risk of contagion and improving the timeliness and continuity of their monitoring activities. This should be the case, especially, for subjects with a chronic condition.

In case of another lockdown, which may further diminish the general health of the population, simple workouts may be proposed in the form of instructional videos and/or online tutorials to people confined in their homes. In addition, personal training may be supplied by means of the remote supervision of fitness professionals, using videoconferencing tools. As one of the studies [26] which showed an increase in PA during lockdown suggested, the main choice for such workouts may be body-weight training (calisthenics), which can be easily performed at home without any equipment but a simple yoga mat. Yoga itself may be a good option, given the positive effects which it may have on inflammatory conditions and on the immune system. 

Another interesting aspect that emerges from the 23 selected studies is linked to the psychology and motivation that drive PA: higher anxiety scores during the COVID-19 lockdown negatively influenced commitment to exercise. This may lead to practical implications when considering the confinement period that people are held at home for long periods of time, since psychological conditions (anxiety and depression in particular) must also be taken into consideration. Unfortunately, motivation to exercise seems to diminish in individuals who are confined at home, and home workouts do not have the same benefits as outdoor PA for the immune system [35]. Exposure to sunlight has proved to have important beneficial effects on the immune system response, including—of course—reaction to viral infections [36]. Subjects without access to a garden, courtyard, or terrace may be at a further disadvantage when confined at home.

This is especially important for older individuals, even in the case of a forthcoming vaccine. Infections have a greater incidence in the elderly [37], and their immune systems respond less effectively to vaccines, while physical exercise improves the effectiveness of the immune system response not only to infections but also to vaccinations [38]. Recent studies have found a positive effect of PA on patients with chronic conditions, including autoimmune diseases [39]. It must be highlighted that COVID-19 may lead to an intense and fatal cytokine response, akin to an autoimmune reaction.

As it is usually the case, any policy or measure which is enforced in order to diminish a risk (including a health-related risk, such as that of a viral infection) has, as a side effect, the potential of increasing another risk. In case of subsequent lockdowns, health-related risks, for both healthy individuals and patients with chronic diseases, may increase because of lack of PA and exposure to sunlight. In a seemingly paradoxical way, even infection-related risks may specifically increase. In fact, those measures which are put in place to diminish the spread of an infection in a population in the short run may hinder the capability of the population to react to the same infection—or even to other kinds of infections—in the long run.

The limitations of the selected studies can be summarized as follows. First, there was no consistency in the tools and methods of assessing PA, making the comparison of results difficult. Second, all studies but one [20] used self-reported questionnaires to assess the performed PA, and asked individuals about their pre-lockdown behavior retrospectively. These facts diminish both the accuracy and objectivity of the assessment. Finally, the selected samples were not representative of the population at a national level.

## 5. Conclusions

Physical activity is a prerequisite for the prevention and treatment of most medical conditions, especially metabolic, cardiovascular, and neurodegenerative diseases. Furthermore, it is an established fact that outdoor exercise, during sunlight hours, has a positive impact on the immune system, which is of great importance in case of a viral infection. It appears instead that during the outbreak of the pandemic, the whole Italian national health service focused solely and exclusively on the risk posed by the SARS-CoV-2 infection. While this is understandable in the short run, given the terrific burden that it put on hospitals and intensive care units, the negative effects that prolonged home confinement have on the global national health should not be neglected.

A general lockdown, imposed repeatedly and maintained in the long period—without giving alternative and viable options for the practice of physical activity, particularly outdoor—may lead to a worsening of the population general health, in both healthy subjects and those with a chronic condition, creating further medical emergencies and an increased burden on the national health service. People suffering from chronic diseases need special attention during a pandemic, and there should be some plan for them during lockdown to reduce the impact on their health. We suggest, therefore, that at least individual outdoor exercise should be allowed and promoted, especially during daylight hours, while maintaining physical distancing in case another lockdown will be enforced for the containment of current and future pandemics.

Further research is needed in order to assess the amount of performed PA in a more standardized and quantifiable way so that study results can be compared. In addition, it should be necessary to have more information about the use of technological tools and personalized and supervised PA, in order to value their effectiveness.

## Figures and Tables

**Figure 1 ijerph-18-06416-f001:**
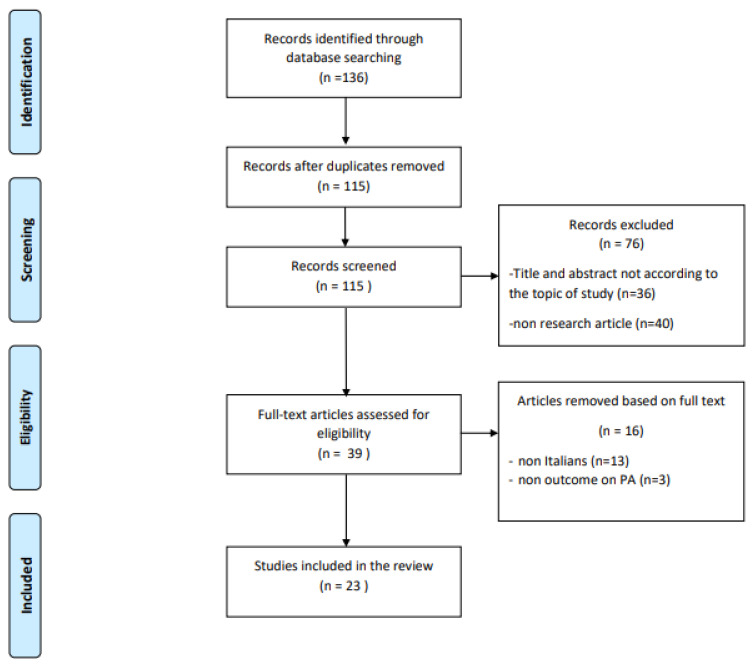
Flow diagram of the literature search strategy and review process, following PRISMA rules [9].

**Table 1 ijerph-18-06416-t001:** Description of the characteristics of the included papers.

Samples with Pathologies	Focus	Study Design	Sample, Pathology, Size, Age, % Males	PA Assessment Tool Survey Type	Amount of PA (Mean ± SD)	Main Findings
Assaloni et al. [12]	PA level in diabetes	Observational, cross-sectional	N = 154 T1D 44.8 ± 12.5 y, 54.5% males	Godin Leisure Time Exercise Q. online survey	Before L: 66 ± 42 min During L: 38 ± 31 min	PA decreased with worst glycaemia
Barchetta et al. [13]	Glycemic control in diabetes	Observational, cross-sectional	N = 50 T1D, 40.7 ± 13.5 y, 62% males	Q not validated Online survey	NR	Reduction of blood glucose control and weekly PA
Capaldo et al. [14]	Glucose control in diabetes	Observational, cross-sectional	N = 207 T1D 38.4 ± 12.7 y, 53.6% males	Q not validated Online survey	NR	Increased glycemic control, more regular meals, reduced PA
Caruso et al. [15]	Glucose control in diabetes	Observational, cross-sectional	N = 48 T1D 42.4 ± 15.9 y, 52.1% males	Q not validated Phone interview	NR	Increased glycemic control, reduced PA
Predieri et al. [16]	Glycemic control in diabetes	Observational, longitudinal	N = 62 T1D 11.1 ± 4.4 y, 50% males	Q not validated telemedicine	Before L: 3.27 ± 2.82 h/w During L: 0.24 ± 0.59 h/w	Decreased PA, improved glycemic control
Tornese et al. [17]	Glycemic control in diabetes	Observational, cross-sectional	N = 13 T1D median age = 14.2 y, 61.5% males	Q not validated telemedicine	During L: 3.3 h/w	Regular PA at home improved glycemic control
Di Stefano et al. [18]	Levels of PA in neuromuscular diseases (NMD)	Observational, cross-sectional	N = 268, 149 NMD patients (57.3 ± 13.7 y, 62.4% males), 119 controls (56 ± 6.8 y, 62.2% males)	IPAQ-SF Phone interview	NMD Before L: 901.3 ± 1299.6 During L: 400.6 ± 1088.5 Controls Before L: 4506.5 ± 7600.1 During L: 2362.3 ± 4498.9	Significantly decrease of PA in both groups
Pellegrini et al. [19]	Changes in weight and diet in obesity	Observational, cross-sectional	N = 150 obese, aged 47.9 ± 16.0, 22% males	Q not validated e-mail		Significant weight increase, reduction in PA
Sassone et al. [20]	Changes in PA in patients with implantable Cardioverter-defibrillators	Observational, cross-sectional	N = 24 cardiac patients, 72 ± 10 y, 70.8% males	Accelerometric sensors- collected data	Before L: 1.6 ± 0.5 h/day During L: 1.2 ± 0.3 h/day	Significant reduction in PA
Schirinzi et al. [21]	Changes in PA in Parkinson disease	Observational, cross-sectional	N = 74 Parkinson disease 61.3 ± 9.3 y, 50% males	IPAQ-SF Online survey	During L: 1994.7 ± 1971 MET-min/w	60% of patients worsened, performing less PA
**Sample without pathologies**	**Focus**	**Study design**	**Sample** **size, age, % males**	**PA assessment tool** **Survey type**	**Amount of PA** **(mean ± SD)**	**Main findings**
Barrea et al. [22]	Sleep quality, Body mass index	Observational, cross-sectional	N = 121 44.9 ± 13.3 y, 35.5% males	Q not validated phone interview	NR	Significant increase in mean body weight and BMI, significant decrease in PA
Buoite Stella et al. [23]	Smart technologies for PA	Observational, cross-sectional	N = 400 35 ± 15 y, 31% males	Online survey IPAQ-SF; daily step count measured by smart devices	Before L: 3101 ± 3815 METs During L: 1839 ± 2254 METs	Significant reduction of performed steps and PA
Cancello et al. [24]	Lifestyle changes during lockdown	Observational, cross-sectional	N = 490 adults 16% males	Q not validated online survey	NR	Reduction of PA in active individuals, inception of PA in sedentary individuals
Di Corrado et al. [25]	Psychological status, PA	Observational, cross-sectional	N = 679 33.4 ± 12.8 y, 51% males	Q not validated Online survey	NR	Maintained or increased PA significantly
Di Renzo et al. [26]	Eating habits and lifestyle changes	Observational, cross-sectional	N = 3533 aged 12–86 23.9% males	Q not validated Online survey	NR	No significant difference in PA among inactive subjects, increase in PA in subjects who used to train more than 5 times a week
Ferrante et al. [27]	Impact of social isolation on lifestyle	Observational, cross-sectional	N = 7847 48.6 ± 13.9 y, 28.7% males	Q not validated Online survey	NR	Significant decrease in PA
Gallè et al. [28]	Sedentary behaviors and PA	Observational, cross-sectional	N = 1430 undergraduate students, 22.9 ± 4.5 y, 34.5% males	IPAQ-SF Online survey	Before L: 520 ± 820 min/w During L: 270 ± 340 min/w	Significantly increased sedentary lifestyle, decreased PA
Gallè et al. [29]	Health-related behaviors PA	Observational, cross-sectional	N = 2125 undergraduate students, 22.5 ± 0.08 y, 37.2% males	Questionnaire online survey	NR	Significant reduction in PA
Giustino et al. [30]	Level of PA	Observational, cross-sectional	N = 802, 32.27 ± 12.81 y, 49% males	IPAQ-SF online survey	Before L: 3006 MET-min/w During L: 1483.8 MET-min/w	Significant reduction of PA, especially in males and in overweight
Luciano et al. [31]	Behaviors during lockdown (PA, sedentariness, sleep)	Observational, cross-sectional	N = 1471 medicine students 23 ± 2 y, 30% males	IPAQ-SF online survey	Before L: 1588 MET-min/w During L: 960 MET-min/w	Decreased PA, and increased sitting and sleep time
Maugeri et al. [32]	PA on psychological	Observational, cross-sectional	N = 2524 43.6% males	IPAQ Online survey	Before L: 2429 MET-min/w During L: 1577 MET-min/w	PA level decreased with negative impact on psychological health
Raiola et al. [33]	Changes in PA	Observational, cross-sectional	N = 268 Mean age = 26 y	Q not validated Online survey	NR	No change in PA
Tornaghi et al. [34]	PA levels	Observational, cross-sectional	N = 1568 students Aged 15–18	IPAQ Online survey	Before L: 1676.37 ± 20.6 MET-min/w After L: 1774.50 ± 33.93 MET-min/w	Inactive or moderately active students unchanged their PA level; highly active ones increased PA level

T1D: Type 1 diabetes; PA: physical activity; IPAQ: International Physical Activity Questionnaire; IPAQ-SF: International Physical Activity Questionnaire Short-Form; NR: not reported; L: lockdown.

**Table 2 ijerph-18-06416-t002:** NOS scores for all included studies (range: 0–10, with higher scores indicating better quality research).

REFERENCE	Representativeness of Sample	Sample Size	Non-Respondents	Ascertainment of the Exposure	Comparability	Assessment of the Outcome	Statistics	NOS Score
Assaloni et al. [12]	1	0	0	2	0	1	1	6
Barchetta et al. [13]	1	0	0	1	0	1	1	4
Barrea et al. [22]	1	1	0	1	0	1	1	5
Buoite Stella et al. [23]	1	0	0	2	0	1	1	5
Cancello et al. [24]	1	0	0	1	0	1	1	4
Capaldo et al. [14]	1	0	0	0	0	1	0	2
Caruso et al. [15]	1	0	0	0	0	1	1	3
Di Corrado et al. [25]	1	0	0	0	0	1	1	3
Di Renzo et al. [26]	1	1	1	1	1	1	1	7
Di Stefano et al. [18]	1	0	0	2	1	1	1	6
Ferrante et al. [27]	1	1	1	1	2	1	1	8
Gallè et al. [28]	1	1	0	2	0	1	1	6
Gallè et al. [29]	1	1	0	2	0	1	1	6
Giustino et al. [30]	1	1	0	1	0	1	1	5
Luciano et al. [31]	1	1	0	2	0	1	1	6
Maugeri et al. [32]	1	0	0	2	0	1	1	5
Pellegrini et al. [19]	1	0	0	1	0	1	1	4
Predieri et al. [16]	1	0	0	1	0	1	1	4
Raiola et al. [33]	1	1	0	1	0	1	1	5
Sassone et al. [20]	1	0	0	2	0	2	1	6
Schirinzi et al. [21]	1	0	0	2	0	1	1	5
Tornaghi et al. [34]	1	0	0	2	1	1	1	6
Tornese et al. [17]	1	0	0	1	1	1	1	5

## Data Availability

Data is contained within the article.

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
