# Peer review of "Physical Activity during COVID-19 Lockdown in Italy: A Systematic Review"

_ijerph, 2021, doi:10.3390/ijerph18126416_

Round 1

Reviewer 1 Report

The paper is current. Your publication is suitable for this magazine. The PRISMA method is correctly applied and provides information to draw conclusions that ratify preconceived ideas in society. Perhaps the authors could be asked to explain the limitations of the study that in turn imply future lines of research. Statistical programs for qualitative techniques could also be used to delve into the 26 selected publications.

Author Response

The paper is current. Your publication is suitable for this magazine. The PRISMA method is correctly applied and provides information to draw conclusions that ratify preconceived ideas in society. Perhaps the authors could be asked to explain the limitations of the study that in turn imply future lines of research. Statistical programs for qualitative techniques could also be used to delve into the 26 selected publications.

A: We would like to express our gratitude to the Reviewer 1 for the useful comments to improve the manuscript. We added the limitations of the study and the results of the Newcastle-Ottawa Scale (NOS) for assessing the quality of the selected publications (Table 2).

Reviewer 2 Report

  • Introduction

The rational of the introduction it’s not exclusive from Italy but is cross wide world. There are reports from different countries that shows the decrease of PA as a consequence of the COVID pandemic. The question is: Were the Italian citizens much more active before the pandemics?

Aim: what kind of review was performed?

  • Material and Methods
  • The authors should justify why they use only the database pubmed and scopus
  • The mesh terms are to board.
  • What were the study design included in the review?
  • Results
  • From the analysis of the table 1, it seems that the reviewers included papers that don’t fits to PA impact. Please clarify.
  • The report of the results should be clearly focused in the worsening of the health conditions and lifestyles
  • What were the other variables included in the studies in order to control the cofounding effect of the association of the PA and the outcomes?
  • What was the technique to analyze the quality of papers?
  • There is no reference on the study design in the results topic. It is important to include this aspect in the table 1
  • Discussion- This topic is to boarder. the authors mix the chronic conditions affected with the reduction and increase in BP levels, methodological issues, even removing extrapolations about the methods used, the type of interventions adopted during confinement and the provision of remote health care. The discussion needs to be rewritten
  • The conclusion should be shorter and focused in the purpose of the review.

Author Response

A: We would like to express our gratitude to the Reviewer 2 for the useful comments to improve the manuscript.

Introduction

R2: The rational of the introduction it’s not exclusive from Italy but is cross wide world. There are reports from different countries that shows the decrease of PA as a consequence of the COVID pandemic. The question is: Were the Italian citizens much more active before the pandemics?

A: As the studies we selected had a general introduction on the occurrence of COVID we preferred to keep the same approach at least in the first paragraph, to give the readers the necessary context. The rest of the introduction focuses on the Italian case. The information regarding the change in PA during lockdown in Italian citizens has been added in results section.

R2: Aim: what kind of review was performed?

A: We performed a systematic review according to the PRISMA protocol. We have specified the type of review in the title, at the end of the Introduction (aim), as per your suggestion.

Material and Methods

R2: The authors should justify why they use only the database pubmed and scopus

A: We decided to use Pubmed and Scopus databases exclusively because Scopus is the database that indexes a larger number of journal than Web of Science and Google Scholar, and PubMed focuses mainly on Medicine and biomedical sciences; also Google Scholar offers results of inconsistent accuracy, according to Falagas et al. in “Falagas ME, Pitsouni EI, Malietzis GA, Pappas G. Comparison of PubMed, Scopus, Web of Science, and Google Scholar: strengths and weaknesses. FASEB J. 2008 Feb;22(2):338-42”.

R2: The mesh terms are to board.

A: Sorry, but we didn’t understand this comment. We hope we addressed this issue in the current revision. We used MeSH for our search, and the number of selected articles was the same.

R2: What were the study design included in the review?

A: We added the study design details in Table 1.

Results

R2: From the analysis of the table 1, it seems that the reviewers included papers that don’t fit to PA impact. Please clarify.

A: Thank you for your comment. We checked the selected studies, and actually found three which do not fit, because they do not address changes in PA levels. We therefore deleted them, as per your suggestion.

R2: The report of the results should be clearly focused on the worsening of the health conditions and lifestyles. What were the other variables included in the studies in order to control the confounding effect of the association of the PA and the outcomes?

A: We started from the hypothesis that PA changed during lockdown and verified if it increased, diminished or remained the same on the basis of the evidence provided by the selected studies.

R2: What was the technique used to analyse the quality of papers?

A: Thank you for your comment. We analysed the quality of papers using the Newcastle-Ottawa Scale (NOS) (Wells GA, Shea B, O’Connell D. The Newcastle-Ottawa scale (NOS) for assessing the quality of nonrandomised studies in meta-analyses. Ottawa, ON: Ottawa Hospital Research Institute, 2009) adapted for cross-sectional studies (Modesti PA, Reboldi G, Cappuccio FP, et al. Panethnic differences in blood pressure in Europe: a systematic review and meta-analysis. PLoS One 2016;11:e0147601). The results of the assessment are reported in Table 2.

R2: There is no reference on the study design in the results topic. It is important to include this aspect in the table 1.

A: Study design details were added to Table 1.

Discussion

This topic is to boarder. the authors mix the chronic conditions affected with the reduction and increase in BP levels, methodological issues, even removing extrapolations about the methods used, the type of interventions adopted during confinement and the provision of remote health care. The discussion needs to be rewritten. The conclusion should be shorter and focused in the purpose of the review.

A: We did not address BP levels. We integrated the discussion according to your suggestions

Reviewer 3 Report

The authors present an interesting article, in line with the theme of the journal.
I would like to make some suggestions to the authors to improve the understanding of the procedures and the practical applications of the study:
In the method, they should define the variables with which the content of the 26 articles has been analyzed, establishing clearer data for each of the results. 
The information on the variable "focus" is correct (Table 1). However, a more detailed analysis of the content of the research is necessary for the text to verify the hypothesis put forward and to find out about other aspects/variables that revolve around them due to their methodological implications.
- How many studies referred to subjects with and without pathology? 
- Within the subjects with pathology, which were the pathologies analyzed in the research (including psychological ones)?
- Was the validity of the measurement instruments collected?
- Is the physical activity of Italians quantified? The IPAQ and IPAQ-SF instruments quantify it, and the other instruments used in the studies, could provide some data?
- How many studies measured change in physical activity in the period of confinement and how many studies looked at the amount of physical activity and its association with health status?

Results: 
To clarify the data, it would be interesting to present a contingency table with % and frequencies of variables of interest. 

Discussion:
This study aimed to review published work on the effects of confinement on physical activity, and the consequences on general health. It is not disputed that physical activity is necessary for metabolic regulation. However, diet also has an important role in the regulation of overweight and diabetes. In the studies reviewed, what role did the researchers give to diet, could you discuss this?

Twenty-five studies were conducted with self-reported data, eight studies used IPAQ or IPAQ-SF, which questionnaires did the rest of the studies use, how valid and reliable were the data collection instruments? Other valid and reliable questionnaires exist in the scientific literature, were any used? You could discuss the amount, intensity, and type of physical activity performed. 

Two studies found an increase in physical activity during the lockdown period, could you discuss this, do the characteristics of the individuals and the lifestyle before the pandemic influence the maintenance or increase of physical activity level? 

Although there are few papers, two, it is important to discuss this increase in physical activity, to make practical suggestions before further general or partial lockouts occur. what has happened in studies in other surrounding countries with similar or different lockout periods?

The authors should comment on the importance of individualization of exercise and professional advice to avoid damage to health, both in healthy subjects and those with pathologies. How many subjects followed a physical activity plan organized by a qualified professional, watched videos on their own, or designed their own physical activity plan? This is important for future research.

Author Response

The authors present an interesting article, in line with the theme of the journal. I would like to make some suggestions to the authors to improve the understanding of the procedures and the practical applications of the study.

A: We would like to express our gratitude to the Reviewer 3 for the useful comments to improve the manuscript.

Methods

Authors should define the variables with which the content of the 26 articles has been analysed, establishing clearer data for each of the results. The information on the variable "focus" is correct (Table 1). However, a more detailed analysis of the content of the research is necessary for the text to verify the hypothesis put forward and to find out about other aspects/variables that revolve around them due to their methodological implications.

A: Thanks for your comments. We analysed the content on the basis of focus, study design, sample description, data collection methods and main findings. These variables have been detailed in Methods. Further, they have been detailed in Table 1.

- How many studies referred to subjects with and without pathology?

- Within the subjects with pathology, which were the pathologies analysed in the research (including psychological ones)?

- Was the validity of the measurement instruments collected?

- Is the physical activity of Italians quantified? The IPAQ and IPAQ-SF instruments quantify it, and the other instruments used in the studies, could provide some data?

- How many studies measured change in physical activity in the period of confinement and how many studies looked at the amount of physical activity and its association with health status?

A: All the above-mentioned information have been added in table 1 and specified in the results section.

Results

To clarify the data, it would be interesting to present a contingency table with % and frequencies of variables of interest.

A: We agree with you, but, given the heterogeneity of the methods, variables and results, it was not possible to summarize the data in a contingency table.

Discussion

This study aimed to review published work on the effects of confinement on physical activity, and the consequences on general health. It is not disputed that physical activity is necessary for metabolic regulation. However, diet also has an important role in the regulation of overweight and diabetes. In the studies reviewed, what role did the researchers give to diet, could you discuss this?

A: The focus of our review was PA changes during lockdown and our search has concerned this topic; anyway we added the information regarding diet in the discussion.

R3: Twenty-five studies were conducted with self-reported data, eight studies used IPAQ or IPAQ-SF, which questionnaires did the rest of the studies use, how valid and reliable were the data collection instruments? Other valid and reliable questionnaires exist in the scientific literature, were any used? You could discuss the amount, intensity, and type of physical activity performed.

A: The discussion regarding the amount intensity and type of PA performed has been added in discussion section.

R3: Two studies found an increase in physical activity during the lockdown period, could you discuss this, do the characteristics of the individuals and the lifestyle before the pandemic influence the maintenance or increase of physical activity level?

Although there are few papers, two, it is important to discuss this increase in physical activity, to make practical suggestions before further general or partial lockouts occur. what has happened in studies in other surrounding countries with similar or different lockout periods?

A: Only two studies reported an increase in PA: one regards people who were highly active also before the lockdown (5 times or more a week), the other reported the percentage of subjects that began PA practice during lockdown, which was greater than that of subjects that stopped to do it.

R3: The authors should comment on the importance of individualization of exercise and professional advice to avoid damage to health, both in healthy subjects and those with pathologies. How many subjects followed a physical activity plan organized by a qualified professional, watched videos on their own, or designed their own physical activity plan? This is important for future research.

A: These topics are out of the scope of the present review, whose aim was only to evaluate an eventual increase or decrease in PA during lockdown. Only one study (Schirinzi et al.) reported the results of technology-based tools to assist physical exercise and did not find any difference between levels of MET between users and the non-users, probably due to the old age of the sample. We do think they can be valid references for further studies, as we put in the conclusion.

Round 2

Reviewer 3 Report

I congratulate the authors on the changes made to the study, as they improve the quality of the systematic review process and present the knowledge gained and limitations of the studies analyzed.